# How does the development of the digital economy in RCEP member countries affect China's cross-border e-commerce exports?

**Dong Wang[1], Peiyuan Xu[1], Bowen An[1]\*, Yingying Song[2]\***

**1** College of Economics and Finance, Huaqiao University, Quanzhou, China, **2** College of Statistics and Mathematics, Hebei University of Economics and Business, Shijiazhuang, China

\* 444097234@qq.com (BA); songyingying2222@163.com (YS)

## Abstract

As a significant new mode of trade export in the digital economy era, cross-border e-commerce injects new momentum into trade cooperation among the Regional Comprehensive Economic Partnership (RCEP) member countries. This paper utilizes multi-dimensional panel data constructed from World Bank databases and UNCTAD databases from 2012 to 2021 to analyze the impact mechanism of RCEP member countries' digital economy development on China's cross-border e-commerce export through direct and indirect channels and conducts empirical tests on it. The research results show that, in terms of direct impact, the development of the digital economy in RCEP member countries has promoted China's cross-border e-commerce export, and its impact is heterogeneous. Regarding indirect impact, improving the quality of RCEP member countries' transportation infrastructure and institutional quality is a moderating factor promoting China's cross-border e-commerce export. This study provides important insights for deepening the development of the digital economy in RCEP countries and promoting China's cross-border e-commerce export.

## 1. Introduction

The rapid rise of the digital economy has attracted widespread attention globally, becoming one of the essential driving forces for today's world economic development. A digital economy is an economic form that revolves around information and communication technology (ICT), utilizing new-generation technologies such as the Internet, big data, and artificial intelligence for production, circulation, transactions, and consumption [1]. With the continuous progress and application of information technology, the digital transformation of the economy has profoundly changed the operational methods across various industries, leading to far-reaching impacts on the international trade landscape and areas such as cross-border e-commerce exports. In the era of the digital economy, information and data have become crucial production factors, and the development of digital technology has driven productivity growth and economic structural adjustments [2–4]. As an integral component of the digital economy, cross-border e-commerce provides new opportunities and tools for enterprises to expand into

**Data Availability Statement:** "The data underlying the results presented in this study are third-party data. Others can access these datasets at the following URLs: International Telecommunications

Union (https://www.itu.int/itu-d/sites/statistics);
World Bank WDI (https://datatopics.worldbank.org/
world-development-indicators); United Nations
Organization for Trade and Development (https://
unctad.org/statistics). The authors confirm that
others can access these data in the same manner
as the authors and that the authors had no special
access privileges to these data."

**Funding:** This work was supported by University-
level Research Project of Xinjiang Institute of
Technology (SQ202407) awarded to BA and XL.

**Competing interests:** The authors have declared
that no competing interests exist.

international markets, reduce transaction costs, and enhance supply chain efficiency [5, 6]. As the world's largest developing country, China has made significant progress in the digital economy in recent years, continuously improving infrastructure such as digital payments, e-commerce platforms, and logistics networks, leading to the continuous expansion of cross-border e-commerce exports [7]. According to relevant statistics from China Customs, in 2022, China's cross-border e-commerce exports were more than three times the imports, with an export growth rate of 4.9%.

International regional economic integration cooperation is deepening continuously. The signing of RCEP marks forming a comprehensive economic partnership in East Asia, covering member countries that account for nearly one-third of the global population and about two-thirds of global GDP, forming one of the world's largest free trade areas. The implementation of RCEP will further promote the development of the digital economy in the region, strengthen trade links and cooperation among member countries, and provide new opportunities and impetus for cross-border e-commerce development in the era of the digital economy. As a core member of the RCEP, China has actively participated in formulating and signing the agreement. Its cross-border e-commerce exports are crucial in promoting international trade and economic cooperation. Therefore, examining the impact of digital economy development in RCEP countries on China's cross-border e-commerce exports is essential for understanding the evolving digital economy landscape within the RCEP region. This study provides valuable insights for enhancing regional digital economy development and formulating cross-border e-commerce policies, holding significant theoretical and practical importance.

## 2. Literature review

The relevant research literature in this article can be summarized into three categories. The first is research on developing the digital economy in RCEP member countries. Various factors, including regional economic integration, digital infrastructure construction, expansion of e-commerce markets, construction of digital financial systems, talent cultivation, and policy support, influence the development of the digital economy in RCEP member countries. Regarding regional economic integration and digital economy development, the signing of the RCEP has promoted regional economic integration, providing member countries with a broader market space and cooperation opportunities for digital economy development [8]. Through regional economic integration cooperation, member countries can share resources, jointly develop technologies, and enhance the innovation and competitiveness of the digital economy [1]. Regional trade liberalization and investment facilitation provide a favorable policy environment for developing the digital economy [9]. Concerning the construction of regional digital infrastructure and the development of ICT, RCEP member countries have made significant investments in digital infrastructure construction, including broadband network coverage, the application of 5G technology, and the construction of cloud computing platforms [10]. These measures improve the level of digital infrastructure and create favorable conditions for the extension and development of the digital industrial chain [11]. The continuous advancement of ICT also drives the innovation and upgrade of digital economy models [12, 13]. Regarding the expansion of e-commerce markets and the upgrading of digital consumption, with the expansion of e-commerce markets in RCEP member countries, the demand for digital services and products among consumers continues to increase [14]. The digitization of consumer shopping behaviors and habits provides a strong impetus for developing the digital economy [15, 16]. Simultaneously, upgrading digital consumption also prompts enterprises to continuously improve product quality and service levels [17], promoting the

high-end and quality development of the digital economy. Concerning the construction of regional digital financial systems and financial technology innovation, RCEP member countries have strengthened the construction of digital financial systems and the innovative application of financial technology [18]. Emerging new financial services, such as mobile payments, digital currencies, and fintech platforms, provide convenience and efficiency improvements for the financial support of the digital economy and cross-border trade payments [19]. Regarding the cultivation of regional digital economy talents and the demand for digital economy talents, RCEP member countries have increased efforts to cultivate and introduce digital economy talents with the vigorous development of the digital economy. The demand for digital economy talents, including artificial intelligence experts, prominent data analysts, and e-commerce operators, continues to grow, becoming a significant force driving the development of the digital economy [20, 21]. Concerning the innovation and entrepreneurship environment and policy support for the regional digital economy, RCEP member countries have strengthened the construction of the innovation and entrepreneurship environment and policy support, encouraging enterprises to increase research and development investment, accelerate technological innovation, and upgrade industries. This provides strong support for the innovative development of the digital economy and promotes the emergence and application of new technologies and formats [22].

The second is the relevant research on the current situation and characteristics of China's cross-border e-commerce exports. Regarding the analysis of the scale and growth trend of cross-border e-commerce exports, some scholars have conducted detailed analyses of the scale and growth trend of China's cross-border e-commerce exports [23–25], including export data for the past few years, annual growth rates, and detailed analyses of exports of different product categories. Particularly, amidst the continuously changing global trade environment, the changes in the proportion and position of China's cross-border e-commerce exports in the global market and its future development trends, potential opportunities, and challenges have received widespread attention [26, 27]. Regarding supply chain advantages and international division of labor patterns, some scholars have explored the supply chain advantages and international division of labor patterns behind China's cross-border e-commerce exports based on an international division of labor perspective [28]. This includes China's position and role in the global value chain and how Chinese enterprises utilize domestic production, technological, and labor cost advantages to achieve efficient operations and competitive advantages in cross-border e-commerce exports [29, 30]. Regarding price competitiveness and quality assurance, current discussions mainly focus on how competitive Chinese products are in international markets and how Chinese cross-border e-commerce enterprises enhance price competitiveness by reducing costs, improving efficiency, and optimizing supply chain management [31]. Discussions also cover the quality and safety standards of China's cross-border e-commerce exports and how Chinese enterprises ensure product quality and maintain consumer trust and market reputation [32]. In terms of the contribution to and impact on economic growth, related studies primarily analyze the contribution and impact of China's cross-border e-commerce exports on economic growth from a macroeconomic perspective [33, 34]. This includes the pull effect of cross-border e-commerce exports on China's economic growth [35, 36], the impact on foreign trade structure and industrial upgrading [37], and the influence of cross-border e-commerce exports on employment and economic structural adjustments [38].

The third aspect concerns the relevant research on the impact of the digital economic development of RCEP member countries on China's cross-border e-commerce exports. Firstly, regarding the export market structure, developing the digital economy among RCEP member countries has blurred market boundaries, accelerating international market integration [39, 40]. Consequently, Chinese cross-border e-commerce export enterprises face a broader market

scope and diverse competitors [30]. The market participants have diversified, encompassing traditional large enterprises, emerging digital platforms, and small e-commerce businesses, impacting market structure and competition dynamics [41]. Concerning the export competition landscape, the development of the digital economy has brought forth more new competitors, intensifying and complicating export market competition [42, 43]. Changes in the competition landscape manifest in product prices and quality, marketing methods, supply chain management, and other aspects [44]. Consequently, competitors' strategies and market positioning adjust, creating new competitive situations [45]. Regarding adjustments to the export and value chains, digital transformation has made the cross-border e-commerce export industry chain more intelligent, flexible, and customized, leading to closer and more efficient supply chains and distribution channels [46, 47]. Simultaneously, regional digital economic development has spurred innovation and upgrading along the value chain [48, 49], altering the position and role of Chinese cross-border e-commerce enterprises in the value chain [50]. This has resulted in the creation of more added value and increased profits.

Overall, existing research has mainly focused on discussing the factors influencing the digital economy development of RCEP member countries, the current status and characteristics of China's cross-border e-commerce exports, and the impact of RCEP member countries' digital economy development on China's cross-border e-commerce exports. These discussions have laid the theoretical foundation for this study. Most studies suggest that the digital economy development of RCEP member countries will positively impact China's cross-border e-commerce exports. However, there is scarce empirical literature testing the mechanism of this impact. Based on this, the article first attempts to theoretically analyze the internal mechanism through which the digital economy development of RCEP member countries affects China's cross-border e-commerce exports and proposes research hypotheses. Subsequently, this study focuses on the 15 member countries of RCEP, selects data from 2012 to 2021, uses the entropy method to measure the level of digital economy development in RCEP countries, and employs panel data empirical testing to examine the impact of RCEP member countries' digital economy development on China's cross-border e-commerce exports. The marginal contributions of the article are mainly threefold: First, it combines the reality of RCEP regional digital economy development to analyze in detail the internal mechanism through which the digital economy development of RCEP member countries affects China's cross-border e-commerce exports. Second, it calculates and analyzes the level of digital economy development in RCEP member countries based on the entropy method. Third, it constructs a trade gravity model to empirically test the impact of RCEP member countries' digital economy development level on China's cross-border e-commerce exports.

## 3. Theoretical analysis

In the era of profound transformation in the information technology field, developing the digital economy among RCEP member countries is crucial for China's cross-border e-commerce exports. This study aims to elucidate the impact of the digital economy development in RCEP member countries on China's cross-border e-commerce exports. It further clarifies how digital economy development in these countries influences China's exports through the quality of transportation infrastructure and institutional quality. The direct impact of digital economy development on China's cross-border e-commerce exports is defined as the direct effect, while the influence mediated through transportation infrastructure and institutional quality is identified as the moderating effect. This paper constructs a framework to delineate the direct and moderating effects of digital economy development in RCEP member countries on China's cross-border e-commerce exports (Fig 1).

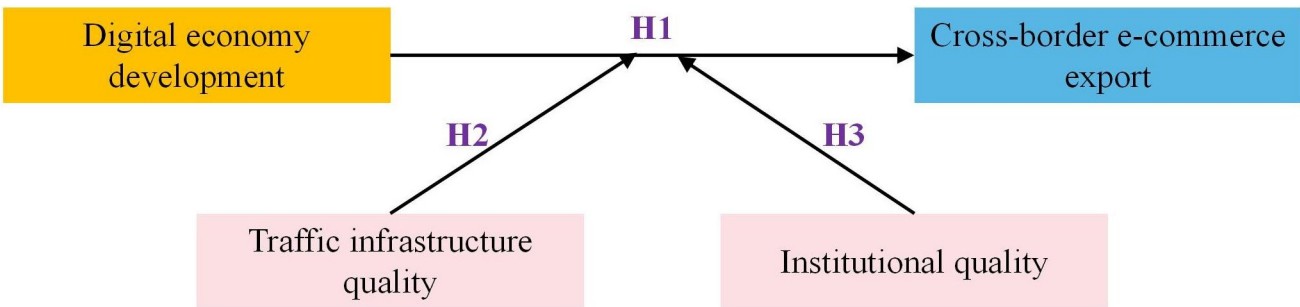

**Fig 1. The influence mechanism of digital economy development in RCEP member countries on cross-border e-commerce exports in China.**

### 3.1 Direct effect

The direct impact of the digital economic development of RCEP member countries on China's cross-border e-commerce exports is multifaceted, covering aspects such as market expansion, enhanced market competitiveness, and supply chain optimization. Firstly, the rapid development of the digital economy among RCEP member countries has expanded market space. As the digitalization level of RCEP member countries improves, the market scope continues to broaden, and consumer demand for Chinese cross-border e-commerce products is increasing. This market expansion effect allows Chinese cross-border e-commerce export enterprises to face a broader market space, conducive to expanding overseas markets and increasing export volume and sales revenue. Secondly, the rapid development of the digital economy among RCEP member countries enhances the market competitiveness of Chinese cross-border e-commerce enterprises. With the widespread application of digital technology, Chinese cross-border e-commerce enterprises can more flexibly utilize technologies such as data analysis, artificial intelligence, and blockchain to optimize product design, market promotion, and customer service, thereby enhancing market competitiveness. The improvement in competitiveness prompts cross-border e-commerce enterprises to continuously improve product quality and innovation levels to meet the constantly changing demands of consumers, thus achieving better performance in a fiercely competitive market. The application of digitalized supply chain management among RCEP member countries makes the supply chain more transparent and efficient, further reducing the costs and risks of cross-border trade. Through the application of digital technology, Chinese cross-border e-commerce enterprises can better achieve real-time monitoring and data analysis of various aspects of the supply chain, improving the flexibility and responsiveness of the supply chain, reducing inventory and transportation costs, and mitigating risks due to uncertainty. The supply chain optimization effect not only enhances the production efficiency of cross-border e-commerce enterprises but also contributes to improving product quality and delivery speed, thus better meeting customer needs and enhancing market competitiveness.

The RCEP consists of 15 member countries, each with varying levels of digital economic development. These differences will have diverse impacts on China's cross-border e-commerce exports. Moreover, each member country's different dimensions of digital economic development, such as digital technology infrastructure, digital talent pool, and digital industry ecosystem, will also lead to differentiated impacts on China's cross-border e-commerce exports. The diversity of these conditions makes the impact of digital economic development in each member country on China's cross-border e-commerce exports varied. Thus, Hypothesis 1 is proposed in this paper.

H1: The direct impact of RCEP member countries' digital economies has facilitated China's cross-border e-commerce exports, and its influence is heterogeneous.

## 3.2 Regulatory effect

**3.2.1 Regulatory effect of traffic infrastructure quality.** Transportation infrastructure refers to the facilities and equipment that enable transportation functions, including roads, railways, air transport, waterways, and urban transport facilities. These structures provide physical channels that connect production factors with markets, facilitating the movement and allocation of resources and benefiting the development of international trade. The quality of transportation infrastructure plays a multifaceted moderating role in the impact of digital economy development in RCEP member countries on China's cross-border e-commerce exports. This includes enhancing logistical efficiency, facilitating the smooth transfer of information, reducing operational costs, and improving the trade environment. Firstly, the quality of transportation infrastructure directly influences the logistical efficiency of cross-border e-commerce. In the era of the digital economy, e-commerce businesses urgently require a fast, efficient, and reliable logistics network, which depends on the seamless transport channels available in RCEP member countries. For instance, high-quality highways, railways, and maritime ports can significantly increase the speed and safety of goods transportation, thereby reducing transaction costs for cross-border e-commerce and enhancing the export competitiveness of Chinese e-commerce to RCEP member countries. Secondly, the quality of transportation infrastructure also affects the speed and fluidity of information transfer in the digital economy. In the e-commerce sector, timely information delivery is crucial for the smooth execution of transactions. Adequate transportation infrastructure ensures uninterrupted communication between cross-border e-commerce businesses, enabling real-time transmission of orders, payments, and other information, facilitating the rapid completion of transactions. Therefore, improvements in the quality of transportation infrastructure in RCEP member countries help accelerate the development of China's cross-border e-commerce exports. The quality of transportation infrastructure directly impacts the operational costs and trade environment of cross-border e-commerce. Well-developed transport facilities can effectively reduce logistics costs and enhance trade convenience, thereby stimulating the growth of cross-border e-commerce. RCEP members with superior transportation infrastructure create a more convenient and efficient trade environment, positively affecting their willingness to enhance international trade cooperation and attracting more Chinese cross-border e-commerce exports to these countries. Hence, this paper proposes Hypothesis 2.

H2: In RCEP, member countries with higher quality transportation infrastructure exhibit a more significant impact on China's cross-border e-commerce exports due to their enhanced digital economy development.

**3.2.2 Regulatory effect of institutional quality.** Institutional quality encompasses multiple dimensions, including legal frameworks, regulatory mechanisms, policies, and governance, which are pivotal for cross-border e-commerce exports in digital economic development. Firstly, high-quality institutions provide a stable legal environment, clear property rights protection, and efficient market mechanisms. These elements foster innovation, enhance efficiency, reduce transaction costs, and strengthen market trust and credibility. Within RCEP member states, there are variations in institutional quality. Some countries possess more robust and efficient systems, while others face significant institutional gaps and uncertainties. The differences in institutional quality among RCEP members directly impact the

development of their digital economies, affecting the conduct of China's cross-border e-commerce exports. There are disparities in the levels of digital economic development among RCEP member countries; some have advanced digital infrastructure, digitalized markets, and a thriving digital economy ecosystem, while others need to catch up. A high-quality institutional environment provides crucial support and safeguards for developing the digital economy. For instance, comprehensive intellectual property protection encourages innovation, transparent legal regulations reduce business risks, and a fair, competitive market environment enhances market dynamism. Therefore, the quality of institutions in RCEP member states plays a critical regulatory role in developing their digital economies, influencing innovation capacity, and impacting China's cross-border e-commerce exports. Lastly, in the digital era, a high-quality institutional environment offers a more stable and predictable legal framework for China's cross-border e-commerce exports, reducing transaction costs and risks and facilitating smoother cross-border trade. Thus, institutional quality plays a significant role in China's e-commerce exports to RCEP member countries, affecting cross-border e-commerce development and international trade growth. In summary, institutional quality significantly modulates the impact of digital economic development in RCEP member countries on China's cross-border e-commerce exports. Accordingly, this paper proposes Hypothesis 3.

H3: In the RCEP, member countries with higher institutional quality exhibit a more significant role in promoting China's cross-border e-commerce exports through digital economic development.

## 4. Construction and measurement of digital economy development level system in RCEP member countries

### 4.1 Construction of index system and weight calculation

As a new engine of global economic growth, the digital economy has garnered significant attention from academia due to its emergence and impact. However, there currently needs to be a unified standard for assessing its level of development. Therefore, this paper will use the definition of the digital economy by the G20 as the basis and combine it with the latest academic research, such as studies by [51–53], to select digital infrastructure, digital economy industry applications, and digital innovation capabilities as primary indicators for assessing the development of the digital economy. These primary indicators will be refined into 11 secondary ones to reflect the digital economy's performance comprehensively. This will enable a more accurate assessment of the comprehensive development level of RCEP member countries in the digital economy. Considering the integrity of available data, missing data will be supplemented using the linear trend prediction method. Table 1 shows the selected indicators and the weight results calculated using the entropy method. The steps of the entropy method are as follows:

Standardize the data to ensure comparability: Assume there are m countries and n evaluation indicators, forming the original data matrix.

$$X_{ij} = \begin{bmatrix} X_{11} & \cdots & X_{1n} \\ \vdots & \ddots & \vdots \\ X_{m1} & \cdots & X_{mn} \end{bmatrix} \tag{1}$$

Since all indicators in this study's measurement system for digital economic development

**Table 1. Evaluation indicators for the digital economy development level of RCEP member countries.**

| | Primary indicators | Secondary indicators | Index weight | data source |
|---|---|---|---|---|
| Development level of digital economy | Digital infrastructure | Number of mobile phone users per 100 people | 0.102 | International Telecommunications Union (ITU) |
| | | Internet user rate | 0.121 | World bank WDI |
| | | Number of secure Internet servers per million people | 0.139 | International Telecommunications Union (ITU) |
| | | Internet broadband per capita | 0.030 | International Telecommunications Union (ITU) |
| | Digital industry application | Proportion of ICT products exports | 0.137 | United Nations Organization for Trade and Development (UNCTAD) |
| | | Proportion of ICT services exports | 0.133 | United Nations Organization for Trade and Development (UNCTAD) |
| | | Proportion of exports of medium and high-tech products | 0.013 | World bank WDI |
| | Digital innovation ability | Enrolment rate of higher education | 0.014 | World bank WDI |
| | | Publication of articles in sci-tech periodicals | 0.105 | World bank WDI |
| | | R&D expenditure as a percentage of GDP | 0.071 | World bank WDI |
| | | Patent application | 0.135 | World bank WDI |

are positive indicators, only the standardization method for positive indicators is presented.

$$D_{ij} = \frac{X_{ij} - \min(X_j)}{\max(X_j) - \min(X_j)} \tag{2}$$

Among them, $X_{ij}$ represents the raw data of a specific indicator for country i in the reporting year, $\min(X_j)$ represents the minimum value of a specific indicator among all statistical countries in the reporting year, $\max(X_j)$ represents the maximum value of a specific indicator among all statistical countries in the reporting year, and $D_{ij}$ represents the processed data of the indicator for country i in the reporting year.

Calculate the P-value:

$$P_{ij} = \frac{D_{ij}}{\sum_{i=1}^{m} D_{ij}} \tag{3}$$

Calculate the information entropy value of each indicator:

$$e_j = -K \sum_{i=1}^{m} P_{ij} \ln(P_{ij}) \tag{4}$$

Calculate the information utility value of each indicator:

$$c_j = 1 - e_j \tag{5}$$

Calculate the weights of each indicator:

$$W_j = \frac{c_j}{\sum_{j=1}^{n} c_j} \tag{6}$$

Calculate the comprehensive score for each country:

$$S_i = \sum_{j=1}^{n} W_j \times P_{ij} (i = 1, 2, \ldots m) \tag{7}$$

## 4.2 Analysis of digital economy development scores and results among RCEP member countries

The data in Table 2 indicates that despite some fluctuations, the overall trend of digital economy development among RCEP member countries is positive. These member countries exhibit significant disparities in digital economy development at the national level, showing evident polarisation. Firstly, Singapore ranks at the top in digital economy development, maintaining a level between 0.52 and 0.75, peaking at 0.751. Conversely, Laos ranks lowest, with digital economy development levels ranging from 0.10 to 0.15, peaking at 0.154. Japan, South Korea, Australia, and New Zealand started with digital economy development levels generally above 0.5, indicating a relatively high level. Japan, South Korea, and Australia experienced minor fluctuations, stabilizing around 0.55 during the statistical period, while New Zealand saw levels around 0.45 in some years. China's digital economy development level ranged from 0.35 to 0.59, showing significant fluctuations. Malaysia, the Philippines, Thailand, Vietnam, Brunei, Indonesia, Cambodia, Myanmar, and Laos generally had lower levels of digital economy development, mostly below 0.4. Specifically, Malaysia's digital economy development level was around 0.35, the Philippines and Thailand were around 0.30, Vietnam, Brunei, and Indonesia were between 0.2 and 0.3, while Cambodia, Myanmar, and Laos remained between 0.10 and 0.15. Analyzing these data reveals a close relationship between digital economy development levels and national economic development. Economically advanced countries tend to have higher levels of digital economy development, while countries with lower economic levels exhibit relatively lower levels.

Based on the time series analysis from 2012 to 2021, Singapore's digital economy has consistently maintained a high level. The significant fluctuation from 2018 to 2019 is particularly noteworthy, where its digital economy development index rose from 0.600 to 0.651. However, in the subsequent years, the growth rate of Singapore's digital economy development has

Table 2. Scores of digital economy development level of RCEP member countries in 2012–2021.

| Country | 2012 | 2013 | 2014 | 2015 | 2016 | 2017 | 2018 | 2019 | 2020 | 2021 |
|---|---|---|---|---|---|---|---|---|---|---|
| Singapore | 0.521 | 0.532 | 0.544 | 0.566 | 0.580 | 0.586 | 0.600 | 0.651 | 0.699 | 0.751 |
| Japan | 0.523 | 0.532 | 0.539 | 0.552 | 0.544 | 0.555 | 0.570 | 0.595 | 0.631 | 0.671 |
| South Korea | 0.502 | 0.512 | 0.524 | 0.534 | 0.537 | 0.545 | 0.540 | 0.593 | 0.613 | 0.641 |
| Australia | 0.503 | 0.504 | 0.506 | 0.505 | 0.512 | 0.529 | 0.542 | 0.555 | 0.568 | 0.584 |
| New Zealand | 0.467 | 0.468 | 0.468 | 0.477 | 0.469 | 0.478 | 0.487 | 0.502 | 0.540 | 0.572 |
| China | 0.363 | 0.363 | 0.384 | 0.407 | 0.438 | 0.467 | 0.484 | 0.515 | 0.540 | 0.584 |
| Malaysia | 0.328 | 0.336 | 0.344 | 0.344 | 0.353 | 0.363 | 0.374 | 0.383 | 0.387 | 0.405 |
| Philippines | 0.278 | 0.314 | 0.314 | 0.323 | 0.335 | 0.344 | 0.334 | 0.345 | 0.363 | 0.386 |
| Thailand | 0.272 | 0.273 | 0.281 | 0.282 | 0.291 | 0.299 | 0.310 | 0.318 | 0.332 | 0.349 |
| Viet Nam | 0.205 | 0.224 | 0.250 | 0.258 | 0.276 | 0.293 | 0.304 | 0.306 | 0.315 | 0.343 |
| Brunei | 0.195 | 0.211 | 0.233 | 0.251 | 0.248 | 0.249 | 0.257 | 0.266 | 0.282 | 0.299 |
| Indonesia | 0.212 | 0.211 | 0.230 | 0.238 | 0.238 | 0.248 | 0.251 | 0.262 | 0.267 | 0.276 |
| Cambodia | 0.136 | 0.157 | 0.148 | 0.155 | 0.156 | 0.164 | 0.163 | 0.161 | 0.178 | 0.195 |
| Myanmar | 0.094 | 0.113 | 0.120 | 0.128 | 0.124 | 0.117 | 0.125 | 0.123 | 0.131 | 0.137 |
| Laos | 0.103 | 0.103 | 0.104 | 0.107 | 0.108 | 0.117 | 0.124 | 0.125 | 0.140 | 0.154 |

shown signs of slowing down. Japan's digital economy experienced a noticeable decline in 2016 but recovered and maintained a stable growth trend since 2017. Concurrently, South Korea, Australia, and New Zealand's digital economies remained relatively stable before 2016, hovering around 0.5, but post-2017, these three countries have witnessed accelerated growth in their digital economies. From 2012 to 2015, significant growth was observed in the digital economy development of China, Malaysia, the Philippines, Thailand, Vietnam, Brunei, Indonesia, Cambodia, Myanmar, and Laos in Southeast Asia, with China exhibiting particularly notable growth. Although there was a brief decline in the digital economies of countries like Brunei and Myanmar in 2016, post-2017, most countries have shown an upward trend in digital economy development. This indicates that developing member countries in the RCEP possess substantial potential for development in the digital economy sector.

## 5. Empirical test on the influence of digital economy development of RCEP member states on cross-border e-commerce export in China

### 5.1 Model setting

The present study employs the gravity trade model as the foundational framework to establish the baseline model for analyzing the impact of RCEP member countries' digital economy development on China's cross-border e-commerce exports, as depicted in Eq (1).

$$LnExp_{j,t} = \beta_0 + \beta_1 LnDei_{j,t} + \beta_2 LnGdp_{j,t} + \beta_3 LnPop_{j,t} + \beta_4 LnOpen_{j,t} + \beta_5 LnTar_{j,t} + \beta_5 LnDis_{j,t}$$
$$+ \mu_i + \mu_t + \varepsilon_{ijt} \tag{8}$$

In the model, j represents the corresponding RCEP member country, and t represents the year. $Exp_{j,t}$ represents China's cross-border e-commerce export volume to country j in year t, $Dei_{j,t}$ represents the country j's Digital Economy Index in year t, $Gdp_{j,t}$ represents country j's economic development level in year t, $Pop_{j,t}$ represents country j's total population in year t, $Open_{j,t}$ represents country j's trade openness in year t, $Tar_{j,t}$ represents country j's tariff level in year t, and $Dis_{j,t}$ represents the geographical distance between country j and China. $u_i$ represents country-specific fixed effects, $u_t$ represents year-specific fixed effects, and $\varepsilon_{ijt}$ represents the random error term.

The impact of the digital economy development in RCEP member countries on China's cross-border e-commerce exports is moderated by differences in the quality of transportation infrastructure and economic institutional environment among member countries. The quality of transportation infrastructure ($Tiq_{j,t}$) and institutional quality ($Inst_{j,t}$) of RCEP member countries are used as moderating variables. The data for transportation infrastructure quality ($Tiq_{j,t}$) is sourced from the Global Competitiveness Report, and the data for institutional quality ($Inst_{j,t}$) is obtained from the World Bank's Worldwide Governance Indicators database. The interaction terms of the two moderating variables with the level of digital economy development in member countries ($Tiq_{j,t} \times Dei_{j,t}$) and ($Inst_{j,t} \times Dei_{j,t}$) are included in the baseline regression to analyze further the moderating effects of transportation infrastructure quality and institutional quality of member countries. The model specifications are shown in Eqs (9) and (10).

$$LnExp_{j,t} = \alpha_0 + \alpha_1 LnDei_{j,t} + \alpha_2 LnTiq_{j,t} + \alpha_3 LnDei_{j,t} \times LnTiq_{j,t} + \alpha_4 LnControl_{j,t} + \mu_i + \mu_t + \varepsilon_{ijt} \tag{9}$$

$$LnExp_{j,t} = \gamma_0 + \gamma_1 LnDei_{j,t} + \gamma_2 LnInst_{j,t} + \gamma_3 LnDei_{j,t} \times LnInst_{j,t} + \gamma_4 LnControl_{j,t} + \mu_i + \mu_t + \varepsilon_{ijt} \tag{10}$$

**Table 3. Descriptive statistical results of each variable.**

| Variable | Number of observations | Mean value | Standard deviation | Minimum | Maximum |
|---|---|---|---|---|---|
| Exp | 140 | 109.432 | 164.927 | 0.755 | 1276.036 |
| Dei | 140 | 0.173 | 0.038 | 0.094 | 0.751 |
| Gdp | 140 | 24106.547 | 2245.951 | 768.365 | 71432.869 |
| Pop | 140 | 5736.146 | 17362.573 | 42.865 | 141208.322 |
| Open | 140 | 0.098 | 0.086 | 0.019 | 0.435 |
| Tar | 140 | 3.116 | 2.377 | 0.023 | 9.698 |
| Dis | 140 | 341657.254 | 243654.935 | 39886 | 1.197E+06 |

## 5.2 Variable selection and data source

The panel data for China's cross-border e-commerce exports to 14 RCEP member countries from 2012 to 2021 are utilized in this study. The variables in the model are categorized into three types: dependent variables, core explanatory variables, and control variables. The dependent variable refers to China's cross-border e-commerce exports to each member country; the explanatory variable is the level of digital economic development of RCEP member countries; and the control variables include a series of important factors that may influence China's cross-border e-commerce exports (see Table 3 for details). The indicators for each category are selected as follows:

The dependent variable. Exp represents the export of China's cross-border e-commerce to each member country each year. Suppose a particular explanatory variable has a positive impact on Exp. In that case, it indicates that the economic activity represented by this explanatory variable has a promotional effect on China's cross-border e-commerce exports. The data for this indicator is sourced from NetEco, and missing data points are supplemented using the linear interpolation method.

The explanatory variables. Dei represents the level of digital economic development for each member country of RCEP each year. This indicator data is calculated as described in the previous sections.

The Control variables. This paper incorporates five control variables: level of economic development (Gdp), population size (Pop), degree of trade openness (Open), tariff rates (Tar), and geographic distance (Dis). The economic development level reflects the economic status of nations, with the GDP of importing countries directly influencing the volume of China's cross-border e-commerce exports. Higher GDP in importing countries suggests more substantial purchasing power, enabling greater capacity to buy goods and services from China. The population size determines the potential market size and demand volume. Larger populations in importing countries likely indicate broader markets, providing expansive opportunities for the growth of Chinese cross-border e-commerce exports. The prominent market sizes typically signify more consumers and potential demand, facilitating the expansion of export volumes for Chinese e-commerce firms. The degree of trade openness illustrates a country's openness to international trade. Importing countries' openness directly impacts the export volume of Chinese cross-border e-commerce. Generally, a more open and facilitative trade environment promotes the development of cross-border e-commerce and an increase in export volumes, whereas restrictive trade practices may limit exports. The ratio of total imports and exports to GDP measures this indicator. Tariff levels in importing countries significantly affect the competitiveness of Chinese goods in those markets. High tariffs increase the market price of Chinese goods, reducing their competitiveness and affecting export volumes. The lower or zero tariffs enhance the price competitiveness of Chinese products, boosting export volumes. All

**Table 4. Correlation test results.**

| Variable | LnExp | LnDei | LnGdp | LnPop | LnOpen | LnTar | LnDis |
|---|---|---|---|---|---|---|---|
| LnExp | - | 0.878*** | 0.713*** | 0.722* | 0.714*** | -0.747** | -0.732*** |
| | | (0.007) | (0.005) | (0.086) | (0.008) | (0.043) | (0.009) |
| LnDei | 0.878*** | - | 0.311** | 0.348*** | 0.324** | -0.339** | -0.303*** |
| | (0.007) | | (0.047) | (0.007) | (0.038) | (0.027) | (0.008) |
| LnGdp | 0.813*** | 0.311** | - | 0.253** | 0.272** | -0.286* | -0.279** |
| | (0.005) | (0.047) | | (0.029) | (0.033) | (0.079) | (0.048) |
| LnPop | 0.522* | 0.348*** | 0.348*** | - | 0.317* | -0.309** | -0.323** |
| | (0.086) | (0.007) | (0.007) | | (0.077) | (0.032) | (0.028) |
| LnOpen | 0.714*** | 0.324** | 0.272** | 0.317* | - | -0.296** | -0.335** |
| | (0.008) | (0.038) | (0.033) | (0.077) | | (0.038) | (0.021) |
| LnTar | -0.747** | -0.339** | -0.286* | -0.309** | -0.296** | - | -0.274*** |
| | (-0.043) | (0.027) | (0.079) | (0.032) | (0.038) | | (0.007) |
| LnDis | -0.832*** | -0.303*** | -0.279** | -0.323** | -0.335** | -0.274*** | - |
| | (0.009) | (0.008) | (0.048) | (0.028) | (0.021) | (0.007) | |

Note: the value of p is in brackets in the table.

"*", "* *" and "* * *" are significant at the significant levels of 10%, 5% and 1% respectively.

these indicators are sourced from the WDI database. Geographic distance primarily reflects the transportation costs of goods; closer geographic proximity generally means lower transportation costs and shorter transit times, making trade interactions with China easier and promoting the growth of cross-border e-commerce exports. In contrast, greater distances can lead to higher transportation costs and longer transit times, potentially reducing the demand for Chinese cross-border e-commerce products and thus limiting export growth. This measure is sourced from the CEPII database.

### 5.3 Empirical analysis and robustness test

**5.3.1 Empirical result analysis.** Before conducting the baseline regression, it is essential to test the variables to ensure their significant impact on China's cross-border e-commerce exports. The findings indicate that all the variables selected in this study significantly correlate with China's cross-border e-commerce exports (Table 4). All variance inflation factor (VIF) values are below 10, indicating that the data meet the multicollinearity criteria (Table 5). Consequently, the selected variables are deemed suitable for baseline regression analysis.

Analysis of benchmark regression results. Based on Eq (8) and panel data from 2012 to 2021 across the sample countries, regression analysis. The Hausman test indicated that a two-

**Table 5. Multicollinearity test results.**

| Variable | VIF | 1/VIF |
|---|---|---|
| LnDei | 2.24 | 0.4464 |
| LnGdp | 4.37 | 0.2288 |
| LnPop | 6.54 | 0.1529 |
| LnOpen | 5.33 | 0.1876 |
| LnTar | 3.06 | 0.3268 |
| LnDis | 4.12 | 0.2427 |
| Mean | VIF | 4.28 |

**Table 6. Model benchmark regression results.**

| Explanatory variable | (1) | (2) |
|---|---|---|
| LnDei | 0.463*** | 0.397*** |
|  | (4.75) | (4.21) |
| LnGdp |  | 0.259** |
|  |  | (2.21) |
| LnPop |  | 0.315* |
|  |  | (1.76) |
| LnOpen |  | 0.413*** |
|  |  | (5.46) |
| LnTar |  | -0.165** |
|  |  | (-2.08) |
| LnDis |  | -0.193*** |
|  |  | (-4.82) |
| Cons | 6.794*** | 7.747** |
|  | (5.24) | (2.25) |
| State fixation effect | Yes | Yes |
| Fixed year effect | Yes | Yes |
| N | 140 | 140 |
| R-Squared | 0.984 | 0.992 |

Note: the value of t is in brackets in the table.

"*", "* *" and "* * *" are significant at the significant levels of 10%, 5% and 1% respectively.

way fixed effects model was appropriate for empirical analysis. This model accounts for fixed effects at both the individual level (e.g., countries or firms) and the time level (e.g., years or quarters), providing a more accurate assessment of the impact of independent variables on the dependent variable while avoiding spurious regression results from unaccounted fixed effects. The regression results are presented in Table 6.

The regression results in Table 6 indicate that the coefficient of the core explanatory variable, namely the level of digital economic development of member countries, is positive and significant. This suggests that developing digital economies in RCEP member countries play a driving role in the growth of China's cross-border e-commerce exports. Additionally, concerning control variables, the regression coefficients of member countries' economic development level, population size, and trade openness are positive and pass the significance test, indicating that these factors positively promote China's cross-border e-commerce exports. This implies that countries with more advanced economies, larger populations, and greater trade openness are more conducive to promoting cross-border e-commerce trade with China. Notably, although the population size factor has a positive impact, its significance could be higher. This may be because the cross-border e-commerce market emphasizes consumption capacity and behavior more than population numbers. Even if a country has a large population, if its consumption capacity is insufficient and market demand is sluggish, the market potential for cross-border e-commerce remains limited. Countries with solid consumption capacity, even with smaller populations, can become significant markets for cross-border e-commerce. Therefore, cross-border e-commerce exports' dependence on the target market's population size factor could be more robust. Furthermore, the regression coefficients of member countries' tariff levels and geographical distance from China are negative and pass the significance test, indicating a negative impact of tariff levels and geographical distance on China's cross-border e-commerce exports. An increase in tariff levels and greater geographical distance

**Table 7. Robustness test results.**

| Explanatory variable | Replacing core explanatory variables | Measure method for changing core explanatory variable | Replace the measurement method |
|---|---|---|---|
| LnDei | | | 0.263** |
| | | | (2.04) |
| LnDer | 0.315** | | |
| | (2.19) | | |
| LnDei_N | | 0.266*** | |
| | | (7.29) | |
| LnGdp | 0.342*** | 0.289** | 0.264*** |
| | (6.34) | (2.23) | (4.28) |
| LnPop | 0.984** | 0.705* | 0.839*** |
| | (2.06) | (1.77) | (7.16) |
| LnOpen | 0.357*** | 0.421*** | 0.392** |
| | (5.73) | (6.45) | (2.23) |
| LnTar | -0.298* | -0.306** | -0.273* |
| | (-1.72) | (-1.98) | (-1.81) |
| LnDis | -0.087* | -0.114* | -0.103** |
| | (-1.75) | (-1.73) | (-2.16) |
| Cons | 11.024** | 9.463*** | 10.371** |
| | (2.01) | (3.95) | (1.99) |
| State fixation effect | Yes | Yes | Yes |
| Fixed year effect | Yes | Yes | Yes |
| N | 140 | 140 | 140 |
| R-Squared | 0.991 | 0.982 | 0.976 |

Note: the value of t is in brackets in the table.

"*", "* *" and "* * *" are significant at the significant levels of 10%, 5% and 1% respectively.

hinder the development of China's cross-border e-commerce exports, reflecting the impact of trade barriers and logistics costs on international trade.

**5.3.2 Robustness test analysis and endogenous discussion.** Replace the core explanatory variables. This study adopts the Network Readiness Index (Der) published by the World Economic Forum as an alternative to the previously calculated comprehensive score of digital economic development. This index consists of three primary indicators: environment, readiness, and usage, which assess the development environment for ICT, the propensity to use ICT, and the actual application of ICT by critical stakeholders. Each primary indicator is subdivided into three secondary indicators, comprising 68 variables. This broad coverage enables a comprehensive evaluation of the digital economic development levels across different countries or regions. Table 7, column (1) presents the results of the re-estimated regression. From the regression results, it is observed that by replacing the score of digital economic development with the Network Readiness Index, the coefficient of the core explanatory variables remains positive and passes the significance test at the 5% level. This reaffirms the reliability of the baseline regression results, demonstrating that the development of the digital economy indeed promotes China's cross-border e-commerce exports.

Replace the core explanatory variable measurement method. The entropy method was previously used to measure the level of digital economic development among RCEP member countries, and based on this, a baseline regression was conducted. In order to avoid potential biases from a single measurement method, this study further utilized principal component analysis to reevaluate the core explanatory variable (Dei_N). It is used as a proxy variable for

the level of digital economic development to verify the robustness of the baseline results. After using different measurement methods, the estimation results are detailed in column (2) of Table 7. The results show that despite using different measurement methods, the impact of digital economic development among RCEP member countries on China's cross-border e-commerce exports remains significant, with the regression coefficient remaining positive. Additionally, the impact of the controlled variables is as expected, further ensuring the reliability and robustness of the baseline analysis results.

Change the measurement method. Outliers in data can significantly impact regression analysis, thus interfering with researchers' accurate understanding of variable relationships. Although the paper has addressed outliers, even after outlier treatment, regression results may still exhibit a certain degree of uncertainty and bias. The study employed quantile regression methods to address this issue and improve result reliability. Quantile regression can better handle the data's nonlinearity, heteroscedasticity, and outlier issues, leading to more reliable regression results. The regression results in Table 7, column (3), validated the previous conclusions through quantile regression, demonstrating that the digital economic development of RCEP member countries significantly promotes China's cross-border e-commerce exports.

Endogenous discussion. When studying the impact of digital economic development on China's cross-border e-commerce exports, it is possible for the level of digital economic development in member countries to also be influenced by cross-border e-commerce exports. This reciprocal causal relationship gives rise to endogeneity issues. This paper employs a lagged processing technique on the core explanatory variables, introducing a one-period delay before conducting tests to eliminate potential interference caused by causal relationships. The study adopts the System Generalized Method of Moments (GMM) to estimate a dynamic panel model. The data presented in Table 8 further supports the previous conclusion that the digital

**Table 8. Estimation results of the GMM method.**

| Explanatory variable | (1) |
|---|---|
| LnDei | 0.427*** |
| | (5.63) |
| LnGdp | 0.605* |
| | (1.83) |
| LnPop | 0.541* |
| | (1.72) |
| LnOpen | 0.720 |
| | (0.99) |
| LnTar | -0.302** |
| | (-2.17) |
| LnDis | -0.224* |
| | (-1.81) |
| Cons | 9.077* |
| | (1.75) |
| State fixation effect | Yes |
| Fixed year effect | Yes |
| AR(2) | 0.248 |
| Hansenp-value | 0.242 |
| N | 140 |

Note: the value of t is in brackets in the table.

"*", "* *" and "* * *" are significant at the significant levels of 10%, 5% and 1% respectively.

economic development of RCEP member countries promotes China's cross-border e-commerce exports.

The estimation using lagged explanatory variables and the System GMM method can partly alleviate endogeneity issues. However, there may still be some things that could be improved in the regression process, such as estimation bias caused by omitted variables or reverse causality. In order to address these potential issues, this study refers to relevant existing research. It uses the fixed telephone subscriptions per 100 inhabitants in RCEP member countries in 2007 as an instrumental variable to measure the level of digital economic development among RCEP member countries. The rise of the digital economy is built upon advancements in internet technology, which can be traced back to the widespread adoption of fixed telephones. The subscription rates of fixed telephones in each member country influence the development of the digital economy by shaping consumer habits and communication technologies, thereby meeting the requirement of instrumentality and correlation with core explanatory variables. Compared to advancements in internet technology and innovations in information technology, the historical impact of fixed telephone quantities on cross-border e-commerce exports is gradually diminishing. Currently, the quantity of fixed telephones also has limited influence on the operations and exports of China's cross-border e-commerce enterprises. Therefore, after controlling for other variables, selecting historical fixed telephone quantities as an instrument variable partly satisfies the homogeneity condition. Based on this rationale, this paper selects the number of fixed telephone subscriptions per 100 people in RCEP member countries in 2007 as an instrument variable to gauge the level of digital economic development across these nations. Following the research methodology outlined in [54], the fixed telephone subscriptions per 100 inhabitants in RCEP member countries in 2007 were used to generate interaction terms with the digital economic development level of RCEP member countries from 2012 to 2021, and the logarithm (LnSls) of these terms was employed as an instrumental variable for each member country's digital economic development level.

Table 9 presents the results of two-stage regression using instrumental variables, with column (1) showing the results of the first-stage regression and column (2) displaying the results of the second-stage regression. In these results, the F-statistics all exceed 10, and the Kleibergen-Paaprk LM statistic, Kleibergen-Paaprk Wald statistic, and F-statistics collectively indicate the validity of the instrumental variables, rejecting the hypotheses of instrumental variable under-identification and weak instrument, thereby demonstrating the rational selection of instrumental variables. The regression results reveal that, after controlling for potential endogeneity issues, the coefficients of digital economic development level are positive and significant, indicating that the digital economic development of RCEP member countries contributes to the growth of China's cross-border e-commerce exports.

## 5.4 Heterogeneity analysis

**5.4.1 Heterogeneity analysis of distinguishing the dimensions of the digital economy.** The impact of digital economy development in different sectors of member countries on China's cross-border e-commerce exports varies to different degrees. In order to gain a deeper understanding of this variation, this paper examines the impact of member countries' digital economy development on China's cross-border e-commerce exports from four dimensions: digital infrastructure, digital industry application, digital innovation capability, and others. The specific regression results are detailed in Table 10.

The regression results from Table 10 reveal that the regression coefficients for digital economic development across various dimensions in RCEP member countries are all positive and statistically significant at the 1% level. Specifically, the coefficient for digital industry

**Table 9. IV-2SLS test results.**

| Explanatory variable | First stage | Second stage |
|---|---|---|
| LnSls | 0.217*** | |
| | (4.55) | |
| LnDei | | 0.195*** |
| | | (8.02) |
| LnGdp | 0.176*** | 0.316*** |
| | (5.68) | (4.65) |
| LnPop | 0.097* | 0.173* |
| | (1.82) | (1.79) |
| LnOpen | 0.239*** | 0.352*** |
| | (3.97) | (5.48) |
| LnTar | -0.134* | -0.228** |
| | (-1.88) | (-2.15) |
| LnDis | -0.141* | -0.209* |
| | (-1.79) | (-1.81) |
| Cons | 6.295** | 7.522* |
| | (2.44) | (1.76) |
| Kleibergen-Paaprk LM | 27.764 | |
| | [0.00] | |
| Kleibergen-Paaprk Wald F | 58.979 | |
| | [16.44] | |
| State fixation effect | Yes | Yes |
| Fixed year effect | Yes | Yes |
| F statistics | 60.43 | |
| N | 140 | 140 |
| R-Squared | 0.978 | 0.989 |

Note: the value of t is in brackets in the table.

"*", "* *" and "* * *" are significant at the significant levels of 10%, 5% and 1% respectively. The value in brackets in [] is the critical value of the Stock-Yogo test at a 10% level.

application is the largest, indicating that developing digital industry applications in RCEP member countries impacts China's cross-border e-commerce exports. This is mainly because the advancement in digital industry applications creates a broader market space for China's cross-border e-commerce. The highly developed digital industry application in member countries implies that China's cross-border e-commerce can offer a more diversified and high-quality product supply, meeting the diverse needs of consumers in member countries, thereby stimulating the growth of China's cross-border e-commerce exports. Next is digital innovation capability. Countries with solid digital innovation capabilities often possess advanced technologies and innovative products, which may have high market competitiveness and attractiveness. For China's cross-border e-commerce, if RCEP member countries exhibit solid digital innovation capabilities, China can leverage these countries' innovative products to expand its cross-border e-commerce product line and enhance export quality and value-added, thereby promoting export growth. The countries with solid digital innovation capabilities typically have a higher level of digital economic policies and regulations, providing a favorable policy environment and legal protection for developing China's cross-border e-commerce, helping to reduce operating costs and risks, and improving export efficiency and quality. Finally, there is digital infrastructure. The higher the level of development of digital infrastructure in member

**Table 10. Regression results of distinguishing digital economy dimensions.**

| Explanatory variable | Digital infrastructure | Digital industry application | Digital innovation ability |
|---|---|---|---|
| LnDei | 0.427*** | 0.583*** | 0.554*** |
| | (6.18) | (5.61) | (6.23) |
| LnGdp | 0.591** | 0.652** | 0.539*** |
| | (2.03) | (1.98) | (3.35) |
| LnPop | 0.772** | 0.834*** | 0.891*** |
| | (2.29) | (3.52) | (4.26) |
| LnOpen | 0.539*** | 0.421*** | 0.468*** |
| | (4.97) | (6.34) | (5.61) |
| LnTar | -0.075 | -0.287* | -0.314* |
| | (-1.19) | (-1.74) | (-1.86) |
| LnDis | -0.223** | -0.316** | -0.295** |
| | (-2.26) | (-1.98) | (-2.04) |
| Cons | 8.487*** | 9.857*** | 8.978*** |
| | (7.55) | (8.48) | (7.87) |
| State fixation effect | Yes | Yes | Yes |
| Fixed year effect | Yes | Yes | Yes |
| N | 140 | 140 | 140 |
| R-Squared | 0.994 | 0.987 | 0.979 |

Note: the value of t is in brackets in the table.

"*", "* *" and "* * *" are significant at the significant levels of 10%, 5% and 1% respectively.

countries, the better the network interconnection, allowing cross-border e-commerce enterprises to process orders and exchange information more quickly, improving logistics efficiency, shortening delivery cycles, and promoting the export of China's cross-border e-commerce products. Furthermore, improving digital infrastructure can enhance the security and reliability of cross-border e-commerce platforms. Secure and stable digital infrastructure helps prevent cybersecurity risks and data breaches, enhancing consumer trust in cross-border e-commerce platforms and driving the sales of China's cross-border e-commerce products in international markets.

**5.4.2 Heterogeneity analysis of differentiating economic development levels.** Different RCEP member countries have differences in the level of digital infrastructure, the breadth of digital industry application, digital innovation capabilities, market demand, and consumer behavior. Therefore, their impact on China's cross-border e-commerce exports also varies. Based on this, this paper distinguishes RCEP member countries into developed and developing countries. Developed countries include Japan, South Korea, Australia, New Zealand, and Singapore, while developing countries include the ASEAN 9 countries excluding Singapore. Empirical tests examine the impact of different countries' digital economic development on China's cross-border e-commerce exports. Specific regression results are detailed in Table 11.

The digital economic development of RCEP member countries at different levels of economic development has all contributed to China's cross-border e-commerce exports. The digital economic development of advanced economies has had a more significant stimulating effect on China's cross-border e-commerce exports. Advanced economies typically possess more sophisticated and advanced digital infrastructure, including high-speed Internet, intelligent logistics systems, and electronic payments. These enhance the operational efficiency and service quality of cross-border e-commerce enterprises, facilitating rapid product circulation and transaction convenience, thus benefiting Chinese cross-border e-commerce export

**Table 11. Regression results of differentiating economic development level.**

| Explanatory variable | Developed countries | Developing country |
|---|---|---|
| LnDei | 0.576*** | 0.461** |
| | (3.43) | (2.17) |
| LnGdp | 0.615* | 0.546** |
| | (1.79) | (2.02) |
| LnPop | 0.984** | 0.742** |
| | (2.06) | (1.98) |
| LnOpen | 1.218*** | 1.015*** |
| | (5.13) | (5.94) |
| LnTar | -0.298 | -0.174 |
| | (-0.97) | (-1.26) |
| LnDis | -0.229* | -0.102* |
| | (-1.82) | (-1.91) |
| Cons | 13.613*** | 8.285*** |
| | (4.35) | (5.37) |
| State fixation effect | Yes | Yes |
| Fixed year effect | Yes | Yes |
| N | 50 | 90 |
| R-Squared | 0.993 | 0.982 |

Note: the value of t is in brackets in the table.

"*", "* *" and "* * *" are significant at the significant levels of 10%, 5% and 1% respectively.

enterprises in market expansion and competitiveness enhancement. Advanced economies have a broader and more mature application of industries in the digital realm, covering various sectors. When Chinese cross-border e-commerce enterprises collaborate with or enter the markets of advanced countries, they gain access to a broader range of high-quality, innovative, and diverse digital products and services, improving their product competitiveness and market adaptability. Thirdly, more robust digital innovation capabilities. Advanced economies possess strong innovation capabilities and technological prowess in the digital economy field, continuously introducing new digital products, services, and business models. This gives Chinese cross-border e-commerce enterprises more choices of innovative products and collaboration opportunities, facilitating market share expansion and enhancing their influence in those countries. Lastly, in terms of market demand and consumer behavior, consumers in advanced economies strongly demand digital products and services, particularly favoring novel and high-quality products. They have a greater demand for products from Chinese cross-border e-commerce enterprises, thus increasing China's cross-border e-commerce exports. This validates hypothesis 1, as mentioned earlier.

**5.4.3 Heterogeneity analysis of differentiating the development level of digital economy.** Due to the significant disparities in digital economic development levels among RCEP member countries, their effects on China's cross-border e-commerce exports vary considerably. An empirical analysis of the relationship between digital economic development levels and China's cross-border e-commerce exports was conducted. Based on previously calculated digital economy development scores, RCEP member countries were categorized into two groups: those with higher levels of digital economic development (Singapore, Japan, South Korea, Australia, and New Zealand) and those with lower levels (the remaining nine RCEP member countries excluding China). The regression results are presented in Table 12.

**Table 12. Regression results of distinguishing the development level of digital economy.**

| Explanatory variable | Countries with high level of digital economy development | Countries with low level of digital economy development |
|---|---|---|
| LnDei | 0.542*** | 0.423** |
|  | (3.05) | (2.27) |
| LnGdp | 0.594** | 0.502** |
|  | (2.09) | (2.15) |
| LnPop | 0.973** | 0.706** |
|  | (2.22) | (2.12) |
| LnOpen | 1.304*** | 1.073*** |
|  | (4.97) | (5.65) |
| LnTar | -0.267 | -0.158 |
|  | (-0.88) | (-1.09) |
| LnDis | -0.234* | -0.111* |
|  | (-1.87) | (-1.93) |
| Cons | 12.764*** | 8.441*** |
|  | (4.86) | (5.23) |
| State fixation effect | Yes | Yes |
| Fixed year effect | Yes | Yes |
| N | 50 | 90 |
| R-Squared | 0.995 | 0.989 |

Note: the value of t is in brackets in the table.

"*", "* *" and "* * *" are significant at the significant levels of 10%, 5% and 1% respectively.

Table 12 shows that digital economy development in RCEP countries positively impacts China's cross-border e-commerce exports, regardless of their development levels. However, the impact is more pronounced in countries with higher levels of digital economic development. This enhanced effect can be attributed to several factors: Countries with advanced digital economies have robust digital infrastructure, including high-speed Internet, sophisticated payment systems, and efficient logistics networks. This infrastructure reduces the costs and risks of cross-border e-commerce transactions, enhancing transaction efficiency and security. Consequently, domestic consumers and businesses are more likely to engage in cross-border e-commerce, improving the market competitiveness of Chinese e-commerce products. Nations with advanced digital economies generally possess stronger technological research and development capabilities and higher technology adoption rates. They rapidly implement and disseminate new technologies such as big data analytics, artificial intelligence, and blockchain. These technologies enhance market targeting, supply chain management, and transaction security, improving the operational efficiency and market responsiveness of Chinese e-commerce enterprises. Consumers in high-digital-economy countries typically have higher digital literacy and greater trust in online transactions. Their advanced digital skills and familiarity with e-commerce platforms reduce barriers for cross-border e-commerce enterprises in market promotion and user education. Higher digital literacy and trust facilitate the acceptance and preference for Chinese cross-border e-commerce platforms and products, thereby expanding market share. The policy environment in countries with advanced digital economies generally supports cross-border e-commerce. These governments often implement favorable policies such as tax incentives, intellectual property protection, and data security regulations. Such measures create a favorable business environment, reducing enterprise operational risks and costs and enhancing market stability. This supportive environment enables

Chinese cross-border e-commerce enterprises to invest and expand their businesses more confidently.

## 5.5 Analysis of the impact of the COVID-19 epidemic

The outbreak of COVID-19 at the end of 2019 has profoundly impacted global supply chains and China's cross-border e-commerce exports. From the supply chain perspective, the pandemic has led to production shutdowns, logistics disruptions, and transportation restrictions in various countries, significantly increasing the instability and uncertainty of supply chains. Concurrently, the pandemic has accelerated the digital transformation of enterprises in RCEP member countries, prompting a greater reliance on online platforms for supply chain management and cross-border e-commerce transactions, thereby driving the rapid development of e-commerce. To investigate how the impact of RCEP member countries' digital economic development on China's cross-border e-commerce exports has changed after the outbreak of COVID-19, this paper introduces a time dummy variable, Time (Time = 0 for 2019 and earlier, Time = 1 for 2020 and later). It generates an interaction term, LnDei×Time, with the digital economic development level of RCEP member countries, Dei, replacing the explanatory variable LnDei in the baseline regression. The specific regression results are detailed in Table 13.

The regression results in Table 13 show that the interaction term coefficient between the level of digital economic development among RCEP member countries and the time dummy variable is positive and more significant than the results from the baseline regression. This indicates that after the outbreak of the COVID-19 pandemic, the development of digital economies in RCEP member countries has had a more significant impact on China's cross-border e-commerce exports, thereby increasing China's dependence on the digital economic infrastructure of RCEP member countries. This change can be attributed to several factors. During

Table 13. Regression results of epidemic impact of COVID-19.

| Explanatory variable | (1) |
|---|---|
| LnDei×Time | 0.485*** |
| | (3.96) |
| LnGdp | 0.254** |
| | (2.12) |
| LnPop | 0.327* |
| | (1.86) |
| LnOpen | 0.424*** |
| | (4.82) |
| LnTar | -0.171** |
| | (-2.16) |
| LnDis | -0.205*** |
| | (-4.27) |
| Cons | 7.663** |
| | (2.39) |
| State fixation effect | Yes |
| Fixed year effect | Yes |
| N | 140 |
| R-Squared | 0.994 |

Note: the value of t is in brackets in the table.

"*", "* *" and "* * *" are significant at the significant levels of 10%, 5% and 1% respectively.

the pandemic, lockdowns and social distancing measures affected consumer shopping habits, leading to a sharp increase in online shopping demand. With traditional offline trade restricted, cross-border e-commerce became a critical channel for consumers to access overseas goods. The development of digital economies in RCEP member countries became crucial in this context, as it facilitated more efficient e-commerce platforms, convenient payment systems, and reliable logistics services, thereby exerting a more significant influence on China's cross-border e-commerce exports. Secondly, the pandemic accelerated global digitalization processes, particularly in the e-commerce sector, fostering unprecedented technological innovation and application. Investments and upgrades in digital infrastructure by RCEP member countries, such as the widespread adoption of 5G networks, the proliferation of electronic payment systems, and the application of intelligent logistics technologies, significantly enhanced the operational efficiency of e-commerce. Improvements in the digital economic infrastructure of RCEP member countries not only reduced transaction costs but enhanced transaction speed and security, strengthening their role in promoting China's cross-border e-commerce exports. Lastly, the pandemic-induced disruptions and restructuring of global supply chains increased reliance on digital supply chain management and cross-border e-commerce platforms among enterprises and consumers worldwide. Enhancing digital economic infrastructure in RCEP member countries facilitated efficient supply chain management and seamless integration, ensuring timely delivery of goods to their destinations. For China's cross-border e-commerce enterprises, this has become a crucial factor in enhancing their international competitiveness and export market share.

## 5.6 Mechanism analysis

According to the theoretical mechanism analysis mentioned earlier, member countries with higher quality transportation infrastructure and institutional quality exhibit a more significant impact on promoting China's cross-border e-commerce exports to these countries. We adopt a moderation effect model and conduct tests on the aforementioned moderating mechanisms to test this moderating effect empirically. Detailed regression results can be found in Table 14. Column (1) represents the regression where the baseline model includes transportation infrastructure quality as a moderating variable. In column (2), the correlation term between the moderating variable and the core explanatory variable is included in the regression model. The research findings indicate that the coefficient of the interaction term between the moderating variable and the core explanatory variable is positive and passes the significance test at the 5% level. This suggests that in RCEP member countries, the promotion effect of higher-quality transportation infrastructure on China's cross-border e-commerce exports is more pronounced. Superior transportation infrastructure helps to reduce the costs and time of cross-border trade, thereby enhancing the efficiency and feasibility of China's cross-border e-commerce exports. This validates hypothesis 2, as mentioned earlier.

The results presented in Table 14 show the regression outcomes when incorporating the quality of institutions as a moderating variable into the baseline model in column (3), including the interaction term between the moderating variable and the core explanatory variable in the model in column (4). columntudy reveals that the interaction term's coefficient is positive and significant at the 5% level when considering moderation's interactive effect. This finding suggests that in RCEP member countries, the promotion effect of developing a digital economy in countries with superior institutional quality on China's cross-border e-commerce exports is more pronounced. This is primarily because a superior institutional environment can provide a more stable, transparent, and orderly business environment, thus promoting the development of the digital economy industry in member countries and subsequently

**Table 14. Regulative mechanism regression results.**

| Explanatory variable | (1) | (2) | (3) | (4) |
|---|---|---|---|---|
| LnDei | 0.414*** | 0.523*** | 0.751** | 0.807*** |
| | (4.85) | (4.38) | (2.23) | (4.64) |
| LnTiq | 0.839*** | 0.916** | | |
| | (3.19) | (2.07) | | |
| LnDei×LnTiq | | 0.165* | | |
| | | (1.84) | | |
| LnInst | | | 0.976*** | 1.124*** |
| | | | (5.81) | (4.79) |
| LnDel×LnInst | | | | 0.275** |
| | | | | (2.14) |
| LnGdp | 0.423*** | 0.465** | 0.235*** | 0.363** |
| | (4.66) | (2.15) | (5.26) | (2.32) |
| LnPop | 0.397* | 0.512* | 0.403** | 0.512* |
| | (1.71) | (1.69) | (1.98) | (1.79) |
| LnOpen | 0.289*** | 0.331** | 0.662* | 0.473** |
| | (6.02) | (2.09) | (1.77) | (2.28) |
| LnTar | -0.176** | -0.227** | -0.305* | -0.447* |
| | (-2.23) | (-2.10) | (-1.73) | (-1.86) |
| LnDis | -0.138* | -0.103** | -0.218* | -0.302** |
| | (-1.82) | (-2.37) | (-1.75) | (2.19) |
| Cons | 8.415*** | 7.548* | 6.683** | 7.136** |
| | (4.14) | (1.86) | (2.05) | (2.24) |
| State fixation effect | Yes | Yes | Yes | Yes |
| Fixed year effect | Yes | Yes | Yes | Yes |
| N | 140 | 140 | 140 | 140 |
| R-Squared | 0.984 | 0.991 | 0.988 | 0.986 |

Note: the value of t is in brackets in the table.

"*", "* *" and "* * *" are significant at the significant levels of 10%, 5% and 1% respectively.

increasing China's cross-border e-commerce exports. This validates the hypothesis 3 proposed earlier.

## 6. Conclusion and suggestion

In the era of digital economy, cross-border e-commerce has become a new driving force for the development of traditional trade. Exploring the impact and mechanism of member countries' digital economy development on China's cross-border e-commerce exports has theoretical and practical significance. Empirical research has found that developing the digital economy in RCEP member countries has promoted China's cross-border e-commerce exports. Furthermore, distinguishing between dimensions of the digital economy and levels of economic development, it was found that the impact of digital economy development in RCEP member countries on China's cross-border e-commerce exports exhibits heterogeneity. Developing digital industry applications in developed member countries impacts China's cross-border e-commerce exports. Examining the regulatory mechanisms shows that member countries of RCEP with higher quality in transportation infrastructure and institutional quality exhibit more significant promotion effects on China's cross-border e-commerce exports

through digital economy development. This paper proposes the following policy recommendations based on the research conclusions above.

Strengthen the coordination of digital economy policies. Promoting coordination of digital economy policies is crucial for China to promote cooperation and development in the digital economy with RCEP member countries. China should strengthen the coordination of digital economy policies by establishing a multilateral dialogue mechanism with RCEP member countries, including high-level dialogues on digital economy policies, consultation conferences, and specialized research institutions for digital economy policies. Secondly, China should promote the establishment of a unified digital economy policy framework and standards within the RCEP region. The framework should cover various aspects of the digital economy, including the construction of digital infrastructure, the legal and regulatory system for the digital economy, and policies for developing digital industries. In terms of digital infrastructure construction, emphasis should be placed on advancing the construction of 5G networks, the application of cloud computing technology, and the development of the Internet of Things. Regarding the legal and regulatory system for the digital economy, efforts should be made to improve relevant laws and regulations to ensure the legitimacy and standardization of digital economy development. Regarding policies for developing digital industries, efforts should be made to promote the coordinated development of the digital economy industry chain and facilitate the formation of digital economy industry clusters.

Deepen digital trade cooperation. China should actively promote digital trade cooperation with RCEP member countries by reducing barriers and costs and enhancing convenience and sustainability, thereby promoting cross-border development in the digital economy and China's exports in cross-border e-commerce. Firstly, China should assist member countries in strengthening the construction of their digital trade platforms, providing an efficient and convenient platform environment for digital trade. This includes developing digital trade processes and customs clearance systems, streamlining digital trade procedures, and improving transaction efficiency. China should actively promote cooperation and development in digital payments and cross-border e-commerce by promoting international interconnectivity of payment systems, reducing the costs and risks of cross-border payments, strengthening cooperation and integration among cross-border e-commerce platforms, facilitating cross-border circulation of goods and services, and enhancing the stability of digital trade. China can also promote innovative development in digital trade, encouraging enterprises to utilize digital technologies for cross-border trade and service output, diversifying forms of digital trade, and improving quality.

Assist other member countries in optimizing transportation infrastructure. The quality of transportation infrastructure in member countries significantly impacts the export performance of Chinese cross-border e-commerce enterprises. China should strengthen policy coordination and strategic alignment with other countries, developing joint development plans to ensure transportation infrastructure construction's overall coherence and continuity. Provide financial support and technical assistance, leveraging China's experience and technological advantages in infrastructure construction to help member countries build and upgrade their transportation infrastructure. China should encourage domestic enterprises to participate in transportation infrastructure projects in RCEP member countries. Public-private partnerships can facilitate the implementation and operational management of these projects. China can engage in multilateral cooperation, utilizing the resources of international organizations and financial institutions to jointly advance the modernization of transportation infrastructure in other RCEP countries, thereby enhancing logistics and trade facilitation within the RCEP region.

Enhance talent cultivation and exchange. Talent in the digital economy is the foundation for innovation and development, crucial for China's cross-border e-commerce exports. China should increase investment in cultivating talent in the digital economy field. This includes establishing specialized courses and degree programs related to the digital economy in higher education institutions to nurture more professionals. China should strengthen talent exchange and cooperation with RCEP member countries. By establishing platforms and collaborative projects for talent exchange, China can facilitate the flow and exchange of talent among different countries, fostering the sharing of experiences and technologies. China can elevate professional talent in the digital economy through shared talent resources and cooperative training. By conducting joint training programs and collaborative research with RCEP member countries, China can collectively address technological and talent bottlenecks in the digital economy sector, thereby enhancing the overall digital economy development in the region.

## Author Contributions

**Conceptualization:** Dong Wang, Peiyuan Xu, Yingying Song.

**Data curation:** Dong Wang, Peiyuan Xu.

**Formal analysis:** Peiyuan Xu.

**Funding acquisition:** Peiyuan Xu, Bowen An, Yingying Song.

**Investigation:** Bowen An, Yingying Song.

**Methodology:** Bowen An.

**Project administration:** Bowen An.

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
