## [Decision Letter · Decision Letter 0]

12 Jul 2024

PONE-D-24-19141How does the development of the digital economy in RCEP member countries affect China's cross-border e-commerce exports?PLOS ONE

Dear Dr. An,

Thank you for submitting your manuscript to PLOS ONE. After careful consideration, we feel that it has merit but does not fully meet PLOS ONE’s publication criteria as it currently stands. Therefore, we invite you to submit a revised version of the manuscript that addresses the points raised during the review process.

We look forward to receiving your revised manuscript.

Kind regards,

Dr Madhabendra Sinha

Academic Editor

PLOS ONE

Journal Requirements:

2. In the online submission form, you indicated that ""The·datasets used and analyzed during the current study available from the corresponding author on reasonable request.""

Additional Editor Comments:

The manuscript needs major revision as per comments given by Reviewer 1 and Reviewer 3.

Reviewers' comments:

Reviewer's Responses to Questions

**Comments to the Author**

1. Is the manuscript technically sound, and do the data support the conclusions?

Reviewer #1: Partly

Reviewer #2: Yes

Reviewer #3: Yes

2. Has the statistical analysis been performed appropriately and rigorously? 

Reviewer #1: Yes

Reviewer #2: Yes

Reviewer #3: Yes

3. Have the authors made all data underlying the findings in their manuscript fully available?

Reviewer #1: No

Reviewer #2: Yes

Reviewer #3: No

4. Is the manuscript presented in an intelligible fashion and written in standard English?

Reviewer #1: Yes

Reviewer #2: Yes

Reviewer #3: Yes

5. Review Comments to the Author

Reviewer #1: The study carries special importance in order to offer some good policy suggestions to policy operators. The authors have gone through rigorous exercise from the empirical front to derive the results.

Reviewer #2: The article demands relevance in the context of the development of digital economy and its effect in facilitating e-commerce across countries. Identification of research gap, formation of hypothesis, selection of methodological technique, discussion of results are found to be appropriate. I recommend for publication of this manuscript.

Reviewer #3: Referee Report on:

How does the development of the digital economy in RCEP member countries affect China's cross-border e-commerce exports?

This paper takes up an interesting issue, relating China's cross-border e-commerce export performances to digitalization of rest of REEP member countries by utilizing multi-dimensional panel data from 2012 to 2021. The research results show that, in terms of direct impact, the development of the digital economy in RCEP member countries has promoted China's cross-border e-commerce export, and its impact is heterogeneous. Regarding indirect impact, improving the quality of RCEP member countries' transportation infrastructure and institutional quality are moderating factors promoting China's e-commerce export.

See comments as attachment.

6. PLOS authors have the option to publish the peer review history of their article (what does this mean?). If published, this will include your full peer review and any attached files.

Reviewer #1: No

Reviewer #2: **Yes: **Sreemanta Sarkar

Reviewer #3: **Yes: **Amit K. Biswas

---

## [Author Response · Author response to Decision Letter 0]

22 Jul 2024

1.Table 1 has constructed index weights for evaluation indicators for the digital economy development level of RCEP member countries using Entropy Method. But the readers can’t get any insight from table 1 how is this done! Even a small note on the mechanism in the form of an appendix could have been better. 

In section 4.1, the calculation steps of the entropy method have been added to explain how the index weights and comprehensive scores for the digital economy development level are calculated.

2.The paper uses “Economic Freedom” as an indicator of institutional quality! But World Bank Group’s databank provides a separate “institutional quality” indicator under the head of “Worldwide Governance Indicators”. If data for RECP member countries are available for the study period, the authors should use that(those) indicator(s), instead of “Economic Freedom” which is more general in nature. 

Based on the reviewers' suggestions, the institutional quality indicators from the World Bank's Worldwide Governance Indicators database have been adopted in section 5.1 to replace the original economic freedom indicators. Corresponding changes have been made to the variables in the model.

3.In table 3, should it be “sample size” or number of observations! The paper uses macro and institutional data and hence nothing is taken as “sample”. 

Based on the reviewers' comments, the "sample size" in Table 3 of section 5.2 has been revised to "Number of observations".

4.In table 4, the results are very good. “Dei”, “Open” and “Dis” are strongly affecting e-com exports but “GDP” and “Tar” are having relatively lesser effects. “Tar” effect is understandable perhaps because tariff rates are pretty low and/or unchanging but why “Pop” is having a weak effect – what could be reasons for that? 

Section 5.3.1 provides further explanation and clarification on the potential reasons for the relatively weak impact of the population size factor (Pop) on China's cross-border e-commerce exports.

5.What is digital economy readiness index (Der)? How is it constructed or what is its data source? 

In section 5.3.2, the Digital Economy Readiness Index has been corrected to the Network Readiness Index. Additionally, an explanation and clarification regarding the data sources and indicator composition of the Network Readiness Index have been provided.

6.In line 560 – 62, it is said: “It uses the fixed telephone subscriptions per 100 inhabitants in RCEP member countries in 2007 as an instrumental variable to measure the level of digital economic development among RCEP member countries.” I wonder whether this is a proper proxy measurement of level of digital economic development among RCEP member countries for the period 2012 – 21. 

In section 5.3.2, the discussion of endogeneity and the rationale and logic for using the 2007 fixed telephone subscriptions per 100 inhabitants in RCEP member countries as an instrumental variable to measure the level of digital economic development in RCEP member countries have been revised and improved.

7.In conclusion at line 767, it is said that “……optimize the domestic digital economy environment.” Assuming it is suggested for China, did the author test how can China’s enhanced digitalization positively affect its own e-com export? Or is it a mere suggestion. I think the conclusion can be properly presented based on the findings of the empirical exercises. 

Based on the reviewers' comments and the empirical research findings of the paper, the third part of the conclusions and recommendations has been revised and improved accordingly.

8.Did the author provide data source for “the quality of transportation infrastructure” and “economic freedom”? I might have missed as couldn’t find!

The data sources for the transportation infrastructure quality indicators and institutional quality indicators have been added to section 5.1.

9. Finally and most importantly, the data period is very interesting! Given that the time period is 2012 – 21 and COVID took place in between, if further data are available till 2023 – a very interesting exercise could have been the pre and post COVID differences with an effort to identify the presence of structural break. Two reasons for that – (a) China and most of the RCEP members were the worst victims of the viral outbreak and (b) post COVID, people were forced to develop a better digital technology in every field of life including trade practices. So it could have been quite interesting to see how did the interplay affect the outcome as the supply chain must have been affected.

In Section 5.5, an analysis of the impact of the COVID-19 pandemic has been added to discuss how the development of the digital economy among RCEP member countries affects the changes in China's cross-border e-commerce exports against the backdrop of the pandemic outbreak. Due to data limitations, the study only extended the research data beyond 2023, focusing primarily on analyzing the impact of changes in China's cross-border e-commerce exports from 2020 to 2021 due to the development of the digital economy among RCEP member countries post the pandemic outbreak.

---

## [Editor Report · Decision Letter 1]

30 Jul 2024

PONE-D-24-19141R1How does the development of the digital economy in RCEP member countries affect China's cross-border e-commerce exports?PLOS ONE

Dear Dr. An,

Thank you for submitting your manuscript to PLOS ONE. After careful consideration, we feel that it has merit but does not fully meet PLOS ONE’s publication criteria as it currently stands. Therefore, we invite you to submit a revised version of the manuscript that addresses the points raised during the review process.

We look forward to receiving your revised manuscript.

Kind regards,

Dr Madhabendra Sinha

Academic Editor

PLOS ONE

Journal Requirements:

Additional Editor Comments :

Reviewer 1's comments are not found to be addressed in the latest revised manuscript. It needs revision again and responses to the Reviewer 1's comments are also needed. Manuscript has to be also revised following the Reviewer 1's comments as Follows:

The authors have done a good job in their capacity. Keeping in mind the scientific contributions of the article I recommend for Major revision.

Reviewer’s Report

The study has endeavoured a good topic of research on the role of the development of the digital economy in RCEP member countries upon China's cross-border e-commerce exports. Having the countries’ dependence towards the digital world in today’s business, the study carries special importance in order to offer some good policy suggestions to the global policy operators. The authors have gone through rigorous exercise from the empirical front to derive the results which are theoretically expected. In reviewing the article I found some areas of concern that are to be handled properly to have good research outputs. The comments and suggestions are given below.

1. First of all, the Abstract should not contain any unidentified terms such as RCEP, though it is a popular term.

2. The theoretical conceptualisation is not sufficient. As the study has contained many explanatory and control variables, I suggest the theoretical underpinning should be such that it can explain the core relationships among the variables capturing all of them. The authors may also use the functional forms of the interrelationships among the variables.

3. The study should categorically explain the motivation and importance of the study. Having a moderate score in e commerce index for China, why the authors have considered China, not even Singapore like countries who have very high index values. It should use updated research articles to find the appropriate research gaps.

4. As the study largely depends on the index score of the digital economy of the member countries, the authors should discuss the method of index computations in detail to ease the readability of the readers from the multi-disciplinary areas.

5. Before going to have the econometric estimations in the panel data format, the authors could have tried for pair-wise correlation analysis to have a primary view on the degree of associations among the variables.

6. As having variations in the index scores among the group of RCEP, the authors may segregate/cluster the countries in two different panels to carry out the similar econometric exercise to have better results for two sets of countries to have proper policy formulations.

Decision: Major revision

---

## [Author Response · Author response to Decision Letter 1]

5 Aug 2024

Reviewer 1

1.First of all, the Abstract should not contain any unidentified terms such as RCEP, though it is a popular term. 

The abstract section has been revised.

2.The theoretical conceptualisation is not sufficient. As the study has contained many explanatory and control variables, I suggest the theoretical underpinning should be such that it can explain the core relationships among the variables capturing all of them. The authors may also use the functional forms of the interrelationships among the variables. 

The research focus of this paper has been clarified before Section 3.1. Additionally, the relationships between the digital economy development of RCEP member countries, China's cross-border e-commerce exports, and the moderating variables (transportation infrastructure quality and institutional quality) have been further elucidated.

3.The study should categorically explain the motivation and importance of the study. Having a moderate score in e commerce index for China, why the authors have considered China, not even Singapore like countries who have very high index values. It should use updated research articles to find the appropriate research gaps. 

The final paragraph of Section 1, Introduction, has been revised and supplemented to clarify the motivation and significance of this study. Although China's digital economy development score is low, as a core member of RCEP, China has actively participated in formulating and signing the agreement. Compared to countries with higher digital economy development scores, China's cross-border e-commerce exports significantly promote international trade and economic cooperation. Therefore, as clarified in the Introduction, this paper selects China as the research subject. Based on the reviewer's suggestions, the latest research literature has also been incorporated into the literature review section.

4.As the study largely depends on the index score of the digital economy of the member countries, the authors should discuss the method of index computations in detail to ease the readability of the readers from the multi-disciplinary areas. 

In section 4.1, the calculation steps of the entropy method have been added to explain how the index weights and comprehensive scores for the digital economy development level are calculated.

5.Before going to have the econometric estimations in the panel data format, the authors could have tried for pair-wise correlation analysis to have a primary view on the degree of associations among the variables.

Section 5.3.1 has been expanded to examine the correlation and multicollinearity of the empirical data. The results of these tests have been analyzed and explained to elucidate the relationships among the data.

6.As having variations in the index scores among the group of RCEP, the authors may segregate/cluster the countries in two different panels to carry out the similar econometric exercise to have better results for two sets of countries to have proper policy formulations.

Section 5.4.3 has been expanded to include an analysis of the heterogeneity in digital economy development levels. Using the previously discussed digital economy development scores, RCEP member countries were categorized into high and low digital economy development groups for empirical testing. The results were compared to explore the differences and analyze the potential reasons for these discrepancies.

Reviewer 2

1.Table 1 has constructed index weights for evaluation indicators for the digital economy development level of RCEP member countries using Entropy Method. But the readers can’t get any insight from table 1 how is this done! Even a small note on the mechanism in the form of an appendix could have been better. 

In section 4.1, the calculation steps of the entropy method have been added to explain how the index weights and comprehensive scores for the digital economy development level are calculated.

2.The paper uses “Economic Freedom” as an indicator of institutional quality! But World Bank Group’s databank provides a separate “institutional quality” indicator under the head of “Worldwide Governance Indicators”. If data for RECP member countries are available for the study period, the authors should use that(those) indicator(s), instead of “Economic Freedom” which is more general in nature. 

The institutional quality indicators from the World Bank's Worldwide Governance Indicators database have been adopted in section 5.1 to replace the original economic freedom indicators. Corresponding changes have been made to the variables in the model.

3.In table 3, should it be “sample size” or number of observations! The paper uses macro and institutional data and hence nothing is taken as “sample”. 

The "sample size" in Table 3 of section 5.2 has been revised to "Number of observations".

4.In table 4, the results are very good. “Dei”, “Open” and “Dis” are strongly affecting e-com exports but “GDP” and “Tar” are having relatively lesser effects. “Tar” effect is understandable perhaps because tariff rates are pretty low and/or unchanging but why “Pop” is having a weak effect – what could be reasons for that? 

Section 5.3.1 provides further explanation and clarification on the potential reasons for the relatively weak impact of the population size factor (Pop) on China's cross-border e-commerce exports.

5.What is digital economy readiness index (Der)? How is it constructed or what is its data source? 

In section 5.3.2, the Digital Economy Readiness Index has been corrected to the Network Readiness Index. Additionally, an explanation and clarification regarding the data sources and indicator composition of the Network Readiness Index have been provided.

6.In line 560 – 62, it is said: “It uses the fixed telephone subscriptions per 100 inhabitants in RCEP member countries in 2007 as an instrumental variable to measure the level of digital economic development among RCEP member countries.” I wonder whether this is a proper proxy measurement of level of digital economic development among RCEP member countries for the period 2012 – 21. 

In section 5.3.2, the discussion of endogeneity and the rationale and logic for using the 2007 fixed telephone subscriptions per 100 inhabitants in RCEP member countries as an instrumental variable to measure the level of digital economic development in RCEP member countries have been revised and improved.

7.In conclusion at line 767, it is said that “……optimize the domestic digital economy environment.” Assuming it is suggested for China, did the author test how can China’s enhanced digitalization positively affect its own e-com export? Or is it a mere suggestion. I think the conclusion can be properly presented based on the findings of the empirical exercises. 

Based on the reviewers' comments and the empirical research findings of the paper, the third part of the conclusions and recommendations has been revised and improved accordingly.

8.Did the author provide data source for “the quality of transportation infrastructure” and “economic freedom”? I might have missed as couldn’t find!

The data sources for the transportation infrastructure quality indicators and institutional quality indicators have been added to section 5.1.

9. Finally and most importantly, the data period is very interesting! Given that the time period is 2012 – 21 and COVID took place in between, if further data are available till 2023 – a very interesting exercise could have been the pre and post COVID differences with an effort to identify the presence of structural break. Two reasons for that – (a) China and most of the RCEP members were the worst victims of the viral outbreak and (b) post COVID, people were forced to develop a better digital technology in every field of life including trade practices. So it could have been quite interesting to see how did the interplay affect the outcome as the supply chain must have been affected.

In Section 5.5, an analysis of the impact of the COVID-19 pandemic has been added to discuss how the development of the digital economy among RCEP member countries affects the changes in China's cross-border e-commerce exports against the backdrop of the pandemic outbreak. Due to data limitations, the study only extended the research data beyond 2023, focusing primarily on analyzing the impact of changes in China's cross-border e-commerce exports from 2020 to 2021 due to the development of the digital economy among RCEP member countries post the pandemic outbreak.

---

## [Editor Report · Decision Letter 2]

27 Aug 2024

PONE-D-24-19141R2How does the development of the digital economy in RCEP member countries affect China's cross-border e-commerce exports?PLOS ONE

Dear Dr. An,

Thank you for submitting your manuscript to PLOS ONE. After careful consideration, we feel that it has merit but does not fully meet PLOS ONE’s publication criteria as it currently stands. Therefore, we invite you to submit a revised version of the manuscript that addresses the points raised during the review process. Please submit your revised manuscript by Oct 11 2024 11:59PM. If you will need more time than this to complete your revisions, please reply to this message or contact the journal office at plosone@plos.org. Please include the following items when submitting your revised manuscript:A rebuttal letter that responds to each point raised by the academic editor and reviewer(s). You should upload this letter as a separate file labeled 'Response to Reviewers'.A marked-up copy of your manuscript that highlights changes made to the original version. You should upload this as a separate file labeled 'Revised Manuscript with Track Changes'.An unmarked version of your revised paper without tracked changes. You should upload this as a separate file labeled 'Manuscript'.If applicable, we recommend that you deposit your laboratory protocols in protocols.io to enhance the reproducibility of your results. Protocols.io assigns your protocol its own identifier (DOI) so that it can be cited independently in the future. For instructions see: https://journals.plos.org/plosone/s/submission-guidelines#loc-laboratory-protocols. Additionally, PLOS ONE offers an option for publishing peer-reviewed Lab Protocol articles, which describe protocols hosted on protocols.io. Read more information on sharing protocols at https://plos.org/protocols?utm_medium=editorial-email&utm_source=authorletters&utm_campaign=protocols.

We look forward to receiving your revised manuscript.

Kind regards,

Madhabendra Sinha, PhD in Economics

Academic Editor

PLOS ONE

Journal Requirements:

Additional Editor Comments:

The article has to be revised again based on the comments given by Reviewer 1, as follows:

The authors have done a good job in their capacity. Keeping in mind the scientific contributions of the article I recommend for Major revision.

Reviewer’s Report

The study has endeavoured a good topic of research on the role of the development of the digital economy in RCEP member countries upon China's cross-border e-commerce exports. Having the countries’ dependence towards the digital world in today’s business, the study carries special importance in order to offer some good policy suggestions to the global policy operators. The authors have gone through rigorous exercise from the empirical front to derive the results which are theoretically expected. In reviewing the article I found some areas of concern that are to be handled properly to have good research outputs. The comments and suggestions are given below.

1. First of all, the Abstract should not contain any unidentified terms such as RCEP, though it is a popular term.

2. The theoretical conceptualisation is not sufficient. As the study has contained many explanatory and control variables, I suggest the theoretical underpinning should be such that it can explain the core relationships among the variables capturing all of them. The authors may also use the functional forms of the interrelationships among the variables.

3. The study should categorically explain the motivation and importance of the study. Having a moderate score in e commerce index for China, why the authors have considered China, not even Singapore like countries who have very high index values. It should use updated research articles to find the appropriate research gaps.

4. As the study largely depends on the index score of the digital economy of the member countries, the authors should discuss the method of index computations in detail to ease the readability of the readers from the multi-disciplinary areas.

5. Before going to have the econometric estimations in the panel data format, the authors could have tried for pair-wise correlation analysis to have a primary view on the degree of associations among the variables.

6. As having variations in the index scores among the group of RCEP, the authors may segregate/cluster the countries in two different panels to carry out the similar econometric exercise to have better results for two sets of countries to have proper policy formulations.

Decision: Major revision

---

## [Author Response · Author response to Decision Letter 2]

28 Aug 2024

Reviewer 1

1.First of all, the Abstract should not contain any unidentified terms such as RCEP, though it is a popular term. 

The abstract section has been revised.

2.The theoretical conceptualisation is not sufficient. As the study has contained many explanatory and control variables, I suggest the theoretical underpinning should be such that it can explain the core relationships among the variables capturing all of them. The authors may also use the functional forms of the interrelationships among the variables. 

The research focus of this paper has been clarified before Section 3.1. Additionally, the relationships between the digital economy development of RCEP member countries, China's cross-border e-commerce exports, and the moderating variables (transportation infrastructure quality and institutional quality) have been further elucidated.

3.The study should categorically explain the motivation and importance of the study. Having a moderate score in e commerce index for China, why the authors have considered China, not even Singapore like countries who have very high index values. It should use updated research articles to find the appropriate research gaps. 

The final paragraph of Section 1, Introduction, has been revised and supplemented to clarify the motivation and significance of this study. Although China's digital economy development score is low, as a core member of RCEP, China has actively participated in formulating and signing the agreement. Compared to countries with higher digital economy development scores, China's cross-border e-commerce exports significantly promote international trade and economic cooperation. Therefore, as clarified in the Introduction, this paper selects China as the research subject. Based on the reviewer's suggestions, the latest research literature has also been incorporated into the literature review section.

4.As the study largely depends on the index score of the digital economy of the member countries, the authors should discuss the method of index computations in detail to ease the readability of the readers from the multi-disciplinary areas. 

In section 4.1, the calculation steps of the entropy method have been added to explain how the index weights and comprehensive scores for the digital economy development level are calculated.

5.Before going to have the econometric estimations in the panel data format, the authors could have tried for pair-wise correlation analysis to have a primary view on the degree of associations among the variables.

Section 5.3.1 has been expanded to examine the correlation and multicollinearity of the empirical data. The results of these tests have been analyzed and explained to elucidate the relationships among the data.

6.As having variations in the index scores among the group of RCEP, the authors may segregate/cluster the countries in two different panels to carry out the similar econometric exercise to have better results for two sets of countries to have proper policy formulations.

Section 5.4.3 has been expanded to include an analysis of the heterogeneity in digital economy development levels. Using the previously discussed digital economy development scores, RCEP member countries were categorized into high and low digital economy development groups for empirical testing. The results were compared to explore the differences and analyze the potential reasons for these discrepancies.

Journal Requirements:

The references were carefully checked and revised.

---

## [Editor Report · Decision Letter 3]

11 Sep 2024

How does the development of the digital economy in RCEP member countries affect China's cross-border e-commerce exports?

PONE-D-24-19141R3

Dear Dr. An,

We’re pleased to inform you that your manuscript has been judged scientifically suitable for publication and will be formally accepted for publication once it meets all outstanding technical requirements.

Kind regards,

Madhabendra Sinha, PhD in Economics

Academic Editor

PLOS ONE

Additional Editor Comments (optional):

The manuscript may be accepted.
---

## [Editor Report · Acceptance letter]

11 Oct 2024

PONE-D-24-19141R3 

PLOS ONE

Dear Dr. An, 

I'm pleased to inform you that your manuscript has been deemed suitable for publication in PLOS ONE. Congratulations! Your manuscript is now being handed over to our production team.

Kind regards, 

on behalf of

Dr. Madhabendra Sinha 

Academic Editor

PLOS ONE